# Research on autonomous obstacle avoidance of mountainous tractors based on semantic neural network and laser SLAM

Ningjie Chang[1], Xianghai Yan[1,2], Bingxin Chen[1], Yiwei Wu[1,2], Liyou Xu[1]*

1 College of Vehicle and Traffic Engineering, Henan University of Science and Technology, Luoyang, Henan, China, 2 State Key Laboratory of Intelligent Agricultural Power Equipment, Luoyang, Henan, China

* xlyou@haust.edu.cn

## Abstract

The accuracy and consistency of obstacle avoidance map construction are poor in complex and changeable dynamic environment. In order to improve the driving safety of mountain tractors in complex mountain environment, an autonomous obstacle avoidance method for mountain tractors based on semantic neural network and laser SLAM was studied. Firstly, the RPLIDAR-A1 lidar sensor is used to realize high-precision scanning of mountain environment and construction of observation model. Then, the observed data is input into the lightweight convolutional neural network, and obstacles in the mountain environment are detected and semantic information is extracted through layer pruning and channel pruning strategies, and key information such as the type, size, and location of obstacles is output. Finally, the 3D map of mountain environment containing semantic information is further constructed by combining laser SLAM technology. A* algorithm is used on the map for global path planning to realize the autonomous obstacle avoidance function of mountain tractors. Experimental results show that this method can accurately detect obstacles in mountain environment, identify steep hillsides, rock piles and densely vegetated areas in mountain environment, plan the shortest and optimal driving path, and flexibly avoid diverse obstacles.

## Introduction

In complex terrain such as mountainous areas, as the main force of agricultural production in mountainous areas [1], tractors often face various natural and man-made obstacles when operating, such as steep hillsides, gullies, rocks, trees, and sudden pedestrians or animals. If these obstacles are not identified in time and effectively avoided, it is very easy to cause accidents such as collision or rollover of the tractor, which seriously threatens the safety of the driver and the machine [2]. The autonomous obstacle avoidance method can perceive the surrounding environment in real

**Data availability statement:** All relevant data are within the manuscript and its Supporting Information files.

**Funding:** Key Research and Development Project of Henan Province, 231111112600; Natural Science Foundation of Henan Province, 242300420370; Training Program for Young Backbone Teachers in Undergraduate Universities in Henan Province, 2024GGJS051; Heluo Youth Talent Support Project, 2024 HLTJ03; Henan Province University Science and Technology Innovation Team Support Program Project, 24IRTSTHN029; Key Research Project Plan for Higher Education Institutions in Henan Province, 25B460004.

**Competing interests:** The authors have declared that no competing interests exist.

time, make autonomous decisions and control the tractor to avoid obstacles, thus significantly reducing accident risk and improving operation safety [3]. Therefore, it is of great significance to study the autonomous obstacle avoidance method of mountain tractors and realize their autonomous navigation, obstacle detection and obstacle avoidance decision-making in complex environments for promoting the development of agricultural mechanization and intelligence in mountain areas [4].

Some shortcomings of autonomous obstacle avoidance methods for mountain tractors not only limit the working efficiency of tractors in complex mountain environments, but also highlight the urgency and necessity of autonomous obstacle avoidance methods. For example, Ozdemir A et al. proposed a GET algorithm for tractor obstacle avoidance [5], introduced the gap based elastic tree (GET) algorithm, obtained environmental data and extracted the position, shape, size and other information of obstacles, built a local environmental model, identified the gaps that the tractor can safely pass through, selected the optimal smooth path, and converted it into a specific driving track, the tractor can avoid obstacles independently. However, this obstacle avoidance method is limited by the ability of its data processing algorithm, which makes it impossible to conduct accurate analysis and judgment quickly after receiving sensor data, thus delaying the generation and implementation of obstacle avoidance decisions. Adamiec Wojcik I et al. proposed a lumped mass autonomous obstacle avoidance model for mountain tractors [6], combined with the dynamic characteristics of bending and longitudinal flexibility, extracted effective environmental information, identified obstacles, classified the identified obstacles, evaluated their impact on tractor driving, and formulated specific driving decisions based on the current state of the tractor and obstacle information. However, this obstacle avoidance method relies on simple infrared sensors. In the complex and changeable mountain environment, its perception ability is often limited, and it is unable to accurately and comprehensively identify the surrounding obstacles. Sezer V et al. proposed a tractor obstacle avoidance method based on the FGM algorithm [7]. Referring to the concept of looking ahead distance (LAD) in the geometric path tracking method, they integrated it into the local planner, defined a dynamic and optimized LAD, which can be automatically adjusted according to the tractor speed to optimize the tracking, obstacle avoidance and driving comfort, and combined the following clearance method (FGM) with the global planner. It is applied to mountain tractors as part of a complete autonomous system. However, this obstacle avoidance method may not be able to accurately identify and effectively deal with obstacles of different heights, shapes and materials, resulting in poor obstacle avoidance effect. Conte D et al. proposed a Bayesian autonomous obstacle avoidance method for mountain tractors [8]. Using Bayesian technology, their intentions were inferred by observing the head posture of people walking with the tractor. The Gaussian fusion method was used to synthesize two independent prediction models to predict the future position of people. When determining the target posture of the tractor, adjustments were made according to the predicted movement of people, With the help of unoccupied distance graph, obstacle avoidance is realized. But this obstacle avoidance method not only increases the workload of the driver, but also reduces the working efficiency and

intelligent level of the tractor.Irshayyid A et al. first reviewed the different traffic scenarios discussed in the literature [9], and then conducted a comprehensive review of DRL technology, such as the state representation method for capturing the interactive dynamics required for safe and efficient merging, and the reward formula for managing key indicators such as safety, efficiency, comfort, and adaptability. The insights from this review can guide future research towards realizing the potential of DRL in complex traffic automation under uncertainty. A dynamic adaptive trajectory planning method based on Bezier curve is proposed. Li H et al. first established a mathematical model of the Bezier curve [10] and analyzed its curve characteristics, which helps to establish the correlation between trajectory control points and vehicles and surrounding obstacles. Secondly, a mathematical function representing the Bezier curve was established, with control points as inputs and lane change control curves as outputs, to achieve lane change trajectory planning for automobiles.

Semantic neural network is a kind of neural network that uses deep learning technology to extract high-level semantic information from data. In the autonomous obstacle avoidance of mountain tractors, semantic neural networks can be used to identify obstacles in the surrounding environment, such as trees, rocks, gullies, etc., and understand their attributes and spatial relationships, which will help tractors make more accurate obstacle avoidance decisions in complex mountain environments. The laser SLAM method is a technology that uses laser radar to sense and locate the environment. In the autonomous obstacle avoidance of mountain tractors, the laser SLAM can build a 3D map of the surrounding environment in real time, and accurately estimate the position and posture of the tractor, providing a wealth of environmental information for the tractor and helping to achieve accurate obstacle avoidance control. Therefore, the research of autonomous obstacle avoidance method for mountain tractors based on semantic neural network and laser SLAM is of great significance for improving the efficiency of mountain operations and ensuring the safety of operations.

## Autonomous obstacle avoidance methods for mountain tractors

### LIDAR observation model

In the actual movement process of the tractor, due to the observation noise and error of the sensor, it is difficult to accurately describe the actual state of the tractor. The sensor data is used to establish the tractor observation model, so as to more accurately simulate the perception process of the tractor to the environment. The RPLIDAR-A1 single line laser radar is used as the main sensor to scan the mountain environment, obtain the surrounding environment data, and accurately understand the mountain environment information through the coordinate conversion process and the establishment of observation model.

(1) Coordinate transformation. The conversion of laser measurement data from the radar coordinate system to the tractor coordinate system in a mountainous environment is shown in Fig 1, where, $R$ denotes the coordinates of the tractor in the mountain environment at the current moment; $L$ denotes the original coordinates of the laser irradiation point. $\chi$ denotes the tractor's facing angle, which is the angle between tractor's current facing with the positive direction of the $x$ axis; $\alpha$ denotes the angle between the laser irradiation point and the radar direction, measured directly by the lidar. $\theta$ represents the included angle between the laser irradiation point and the line of origin and the positive direction of the x-axis after coordinate conversion; $d$ represents the straight line distance between the center of the tractor and the laser irradiation point, which is also measured directly by the LiDAR.

Using the trigonometric relationship, the coordinates of the laser irradiation point in the coordinate system of the tractor are calculated and expressed by the formula:

$$x_L = x + d\cos(\chi - \theta) \tag{1}$$

$$y_L = y + d\sin(\chi - \theta) \tag{2}$$

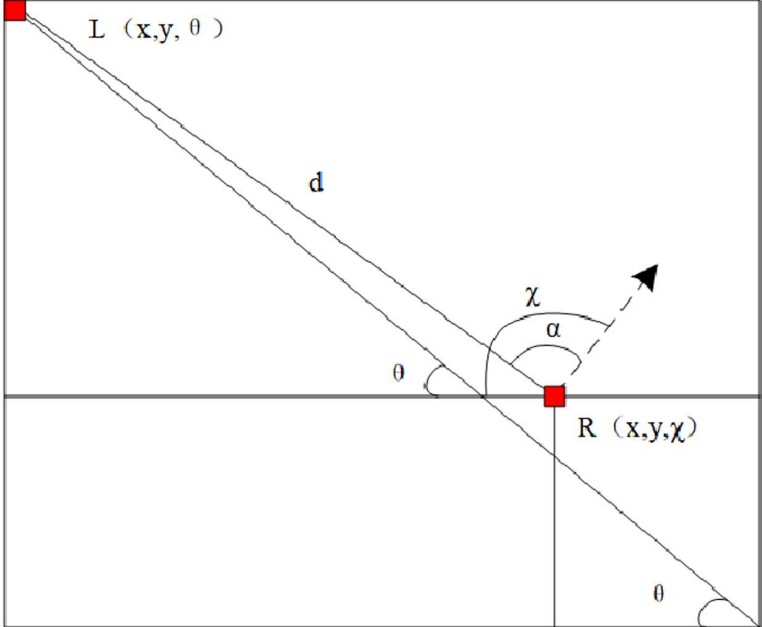

**Fig 1. Coordinate Conversion Process.**

$$\beta = \arctan \frac{y_L}{x_L}$$

[3]

(2) Observational model. During the movement of a tractor in a mountainous environment [11], LIDAR observation will be affected by a variety of factors, including the measurement error and the tractor's movement state. Therefore, it is necessary to establish an observation model that integrates these factors. The observation model is usually expressed as follows:

$$z(t) = \delta(s(t), \phi(t)) = \begin{pmatrix} \sqrt{(x_i - x_t)^2 + (y_i - y_t)^2} \\ \arctan \frac{y_i - y_t}{x_i - x_t} - \theta_t \end{pmatrix} + \phi(t)\beta$$

[4]

Among them, $z(t)$ indicates the number of observations for the radar at time $t$, usually including distance and angle information. $\delta$ denotes the systematic measurement function, it maps the system state $s(t)$ of the tractor and the observation noise $\phi(t)$ at time $t$ into the observation space. $(x_t, y_t)$ denotes the coordinates of the tractor at the moment; $\theta_t$ denotes the angle of orientation of the tractor at the moment; $(x_i, y_i)$ indicates the coordinates of an obstacle in a mountainous environment.

### Neural network-based semantic information extraction for obstacles

Taking the observations $z(t)$ obtained from the observational model described above as the input data of neural network, semantic information of obstacles is extracted. With the pruned YOLOv3 lightweight convolutional neural network and the addition of semantic tags, the accurate semantic information extraction and segmentation of roads, trees, rocks, buildings and other obstacles in mountain environment are realized.

## Design of a lightweight convolutional neural network

Some convolution layers and convolution cores in YOLOv3 network have little impact, even have no impact on the whole network [12]. Therefore, YOLOv3 after pruning is used to detect obstacles in mountain environment observed in LIDAR observation model. Prune the small and ineffective parts of the network. After pruning, YOLOv3's model size and parameters are greatly reduced while maintaining a certain precision.

In the convolutional neural network pruning method [13], layer pruning is based on the role of each convolutional layer as an evaluation criterion, the less influential convolutional layer is cut off, so as to achieve the effect of model compression; channel pruning is to operate on the internal channels or channel weights of the convolutional kernel, and after the channel pruning, the complete convolutional structure will still exist in the network.

During the channel pruning process, the mountain environment feature map matrices and convolution kernel sizes observed in the $i+1$ convolutional layers are respectively $x_{i+1} \in R^{w_{i+1} \times h_{i+1} \times n_{i+1}}$ and $k_{i+1} \times k_{i=1} \times n_{i+1}$, from which the convolution kernel matrix for this convolutional layer can be computed as:

$$Q_{i+1} \in z\left(t\right) R^{n_{i+1} \times k_{i+1} \times k_{i+1} \times n_{j+1}}$$

[5]

In this convolutional layer, if the weight of a channel plays a smaller role in the convolutional calculation, the channel is chosen to be pruned. During convolutional computation, the number of convolutional layers is equal to the number of input feature map channels, so after channel pruning, to adjust the number of channels of the input feature maps of the $x_{i+1}$ convolutional layers. In the previous convolutional layer, a single convolutional kernel can get the corresponding feature map channel in the next convolutional layer after convolution operation, so the corresponding convolutional kernel in the previous layer needs to be clipped.

In the pruning process of lightweight neural network [14], in order to get a large model with high accuracy, normal basic training is needed, and then through sparse training, the unimportant part of the convolutional neural network structure is highlighted and cut out this part, because the accuracy of the previous model will be degraded to varying degrees after the pruning, fine-tuning is needed to restore the model accuracy.

Layer pruning and channel pruning are two methods that have a common feature, they are both structured pruning methods; after pruning, they do not need additional methods to accelerate them. After layer pruning, the depth of the network structure will become shallow. After channel pruning, the width of the network structure becomes narrower. By combining these two methods and pruning the network model together, the depth and width of the network structure will be reduced, and the model can be further compressed by combining the two methods compared to a single pruning method. Therefore, by combining layer pruning and channel pruning methods, a more compact model can be obtained after pruning.

Sparse training is the first step in the pruning process. Through training, some parameters in the model become less important, which makes it easier to remove them when pruning. In YOLOv3, since each convolution layer is followed by a BN (batch normalization) layer, the trainable scale factor $\varphi$ in the BN layer is used as an indicator of channel importance. BN layer uses small batch statistics to normalize convolution characteristics, and the formula is as follows:

$$y = \varphi \times \frac{x - \mu}{\sqrt{\sigma^2 + c}} + eQ_{i+1}$$

[6]

Among them, $x$ denotes the feature map of the output of the convolutional layer; $\mu$ denotes the mean of the feature map; $\sigma^2$ denotes the variance of the feature map; $c$ denotes a positive number in order to prevent the divisor from being zero. $e$ denotes deviation.

In order to effectively differentiate between important and unimportant channels, by giving the scale factor $\varphi$ apply L1 regularization to perform channel sparse training. The training objective formula of sparse training can be expressed as:

$$L = yLoss_{yolo} + \gamma \sum_{\iota \in \Gamma} f(\iota)$$

[7]

Among them, $f(\iota)$ represents L1 norm; $\Gamma$ represents the set of the scale factor $\varphi$ in all BN layers; $\gamma$ denotes the penalty factor that balances the two loss terms. The non-smooth penalty term is optimized using the sub gradient method.

After sparse training, a global threshold $\lambda$ is introduced, based on a global threshold $\lambda$ determine whether a channel should be trimmed. In order to control the trimming ratio of a channel, the global threshold $\lambda$ is set as the $n$ th percentile to all $|\iota|$. In order to prevent over-pruning on the convolutional layer and to maintain the integrity of the network connections, in addition to the global threshold, a local safety threshold $\vartheta$ is introduced. Local security thresholds are set as the $k$ th percentile in a hierarchical manner to the specific layer in $|\iota|$. In pruning, the feature channel whose pruning scaling factor is smaller than the global threshold $\lambda$ and the safety threshold $\vartheta$ minimum. In addition, during the pruning process, the largest pooling and up-sampling layers, which have no relationship with the number of channels, are directly discarded.

Based on the global thresholds $\lambda$ and local safety thresholds $\vartheta$, find the mask (convolution kernel) of all convolution layers. For the routing layer, connect the pruning mask of its incoming layer in order, and use the connected mask as its pruning mask. Traverse the pruning masks of all connected layers, and perform the "OR" operation on these pruning masks to generate the final pruning masks of these connected layers. After pruning, the accuracy of the model will decline to some extent, so it is necessary to train the model again to restore the accuracy of the model.

## Adding semantic tags

In the driving environment of mountain tractors [15], semantic segmentation using the modified YOLOv3 network is a key step to extract semantic information of obstacles. The goal of semantic segmentation is to assign a category label to each pixel in the observed data, including roads, trees, rocks, buildings, etc. in the mountain environment. The model can accurately recognize and segment the obstacles in the mountain environment by precisely labeling the obstacles in the image with the category and boundary information, and inputting them into the lightweight convolutional neural network for training. After the model training is completed, it is applied to the real-time mountain environment image. The model will output the category label of each pixel to achieve the semantic segmentation of obstacles.

The process of adding semantic tags is shown in Fig 2.

The update of semantic tags is divided into three parts, one is the obstacle detection module, the other is the distance calculation module, and the last part is the Marker addition module. When adding semantic tags, the obstacle detection module is responsible for detecting the target obstacles, substituting the color image into the recognition module to obtain the semantic information of the target obstacles and the position of the center point of the bounding box in the color image. The distance calculation module calculates the position of the center point of the boundary box in the depth map to obtain the relative position of the obstacles. According to the coordinate conversion, the global position of the obstacles is obtained. The Marker addition module obtains the semantic tag of the target obstacle by subscribing to the semantic information and global position of the target obstacle.

## Construction of mountain environment map based on laser SLAM

Through the laser SLAM technology, based on the semantic information of the obstacles extracted in Neural network-based semantic information extraction for obstacles, combined with the edge and plane features of the mountain environment extracted by the laser radar scanning, the tractor pose change is estimated using the point-to-point constraint relationship, and the pose and point cloud data are updated iteratively through the graph optimization method, finally the global mountain environment point cloud map is constructed.

Firstly, remove outliers in the point cloud that are too far from the mean. $d_i = \|p_i - \mu\|$, Among them, $p_i$ is the $i$ -th point in the point cloud, and $\mu$ is the value of the point cloud. If $d_i > \tau$ (threshold), it is removed and changed to, and the data is

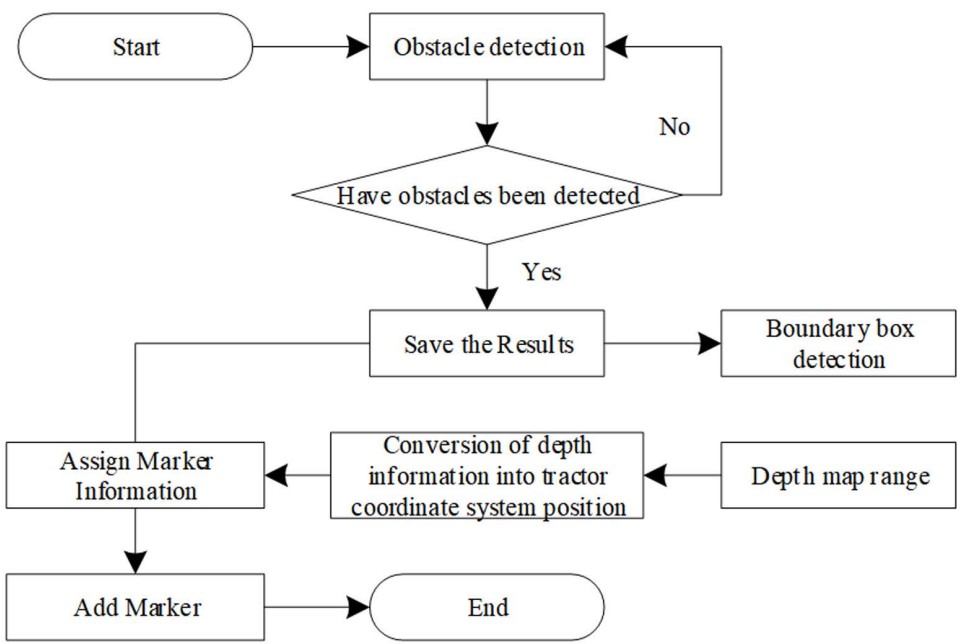

**Fig 2. Semantic Label Addition Process.**

denoised according to the above point cloud filtering. Assuming that the $k$ -th scan of the processed LiDAR extracts a set of edge points in the mountainous environment as $D_k$, Suppose the set of edge points extracted in the $k$ th scan of lidar is $D_k$, and the set of plane points extracted is denoted as $F_k$, the sets of edge point in the mountains environment extracted by the $k+1$ th lidar scan is $D_{k+1}$, and the sets of flat point in the extracted areas is $F_{k+1}$. The point cloud transformation relation between two moments is changed to find the edge feature correspondence and plane feature correspondence between two moments respectively. Pick a point $i$ in $D_{k+1}$, taking the nearest point $j$ to $i$ in $D_k$, and the nearest point $l$ in the scanning line bundle adjacent to $j$ in $D_k$, three points are obtained, the coordinates of which are noted as $x(k,i)$、 $x(k,j)$、 $x(k,l)$, the constraint formula for the edge features can be expressed as follows:

$$d_d = \frac{\left|(x(k,j) - x(k-1,j)) \times (x(k,i) - x(k,l))\right|}{L \left|x(k,i) - x(k,l)\right|}$$

[8]

Among them, $d_d$ denotes the distance between pairs of matching feature points in the set of edge points.

Looking for a point $i$ in $F_{k+1}$, find the point I closest to the point $i$ in $F_k$, find the nearest point $j$ of the laser scanning line beam adjacent to the point $l$ from Hk, and then find the point $m$ closest to the point $j$ in the adjacent frame, four points are obtained, the coordinates of which are noted as $x(k,i)$、 $x(k-1,j)$、 $x(k,l)$、 $x(l,m)$. The constraint formula for a planar feature can be expressed as follows:

$$d_f = \frac{\left|(x(k+1,i) - x(k,j)) \times (x(k,j) - x(k,l)) \times (x(k+1,i) - x(k-1,l))\right|}{\left|x(k,j) - x(k,l)\right|}$$

[9]

Among them, $d_f$ denotes the distance between pairs of matching feature points in a planar point set.

The output of the pose solution part of the laser SLAM problem is the pose information of the tractor, which includes the translation vector $p_w$ and Euler angles $q_w$. $\theta_r$、 $\theta_p$ and $\theta_y$ indicate roll, pitch and heading angles,

respectively. $\varsigma_x$、 $\varsigma_y$ and $\varsigma_z$ are the displacement in the direction $x$、 $y$ and $z$, six-degree-of-freedom variables is denoted by $T_w$.

Using the constraint relation of the point to the surface, the variation $[\varsigma_z, \theta_r, \theta_p]$ of the vertical dimension of the tractor is estimated by matching the correspondence of the plane features extracted into $\{F_k, F_{k+1}\}$ by the lidar scan, matching the edge features in $\{D_k, D_{k+1}\}$ estimates the variations of the level dimension $[\varsigma_x, \varsigma_y, \theta_y]$, obtaining the nonlinear function formula as:

$$f(T^t_{k+1}) = d_f + d_d \qquad [10]$$

The common Levenberg-Marquardt method is used to optimize this nonlinear function to find the minimum distance transformation between two successive scans. Assuming that the objective function is $A(x)$, the error function is given by $\psi(x)$, the least squares optimization of the objective function can be expressed as follows:

$$A(x) = \frac{1}{2} \|\psi(x)\|^2_2 f(T^t_{k+1}) \qquad [11]$$

Since the objective function is difficult to derive, it is not possible to derive the optimal solution of $x$, therefore an iterative method is used to find the incremental $\triangle x_k$, making the error $\|\psi(x_k + \triangle x_k)\|^2_2$ reach a minimal value. Given an initial value $x_0$, for the $k$ th iteration, set the trust region, in the fulfillment of $\|\tau \triangle x_k)\|^2 \leq B$ finding the minimum value, the formula is expressed as follows:

$$\min \frac{1}{2} \left\| \psi(x_k) + O(x_k)^t \triangle x_k) \right\|^2 \qquad [12]$$

Where, $B$ denotes the radius of confidence. $\tau$ denotes the matrix of coefficients; $O$ denotes the initial Jacobi matrix.

Solve the incremental equation by simplifying the derivatives with the Lagrangian function, and the objective function $(T^{L\psi}_{k+1}) = d$ is substituting into the incremental equation, can obtain:

$$-O\psi = (OO^T + \varpi\tau^T\tau) \triangle T \qquad [13]$$

$$\triangle T = -(OO^T + \varpi\tau^T\tau) - O\psi d \qquad [14]$$

Among them, $\varpi$ denotes the Lagrange multiplier.

By continuously iterating the incremental equations, update the position estimates $T^L_{k+1}$.

$$O = \partial\psi \triangle T / \partial T^L_{k+1} \qquad [15]$$

Figure optimized laser SLAM includes two structures: node and edge, which correspond to the position and attitude of the tractor and their relative transformation relationship [16]. Image optimization laser SLAM obtains point cloud data for each frame through laser radar scanning, matches two consecutive frames of point cloud data, and calculates the tractor pose. Fig 3 shows the structure diagram of the graph optimized laser SLAM algorithm, where different waypoints can be observed when the tractor obtains different poses as the input changes [17]. The waypoints obtained at different times can constitute the pose estimation results of mountainous environments. Nodes $v_i$ and $g_i$ in the figure are represented as the position information of the LiDAR (tractor) and the information of the observed road marking points, respectively. The edges of position node-position node and position node-feature node in the figure represent the motion constraints and observation constraints of the LiDAR, respectively.

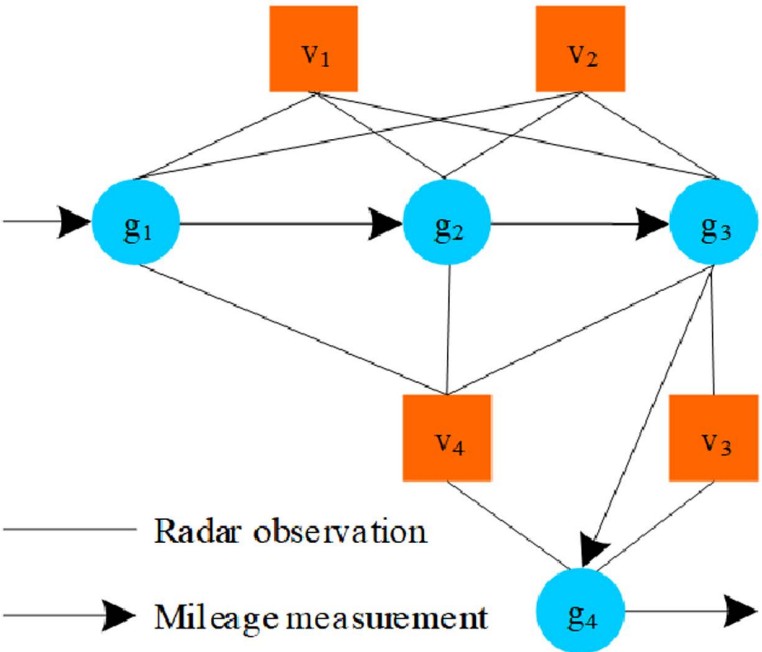

**Fig 3. Structural diagram of optimized laser SLAM algorithm.**

The side from the pose node to the feature node is the observation side, which can be obtained from the observation equation of the laser radar. The edge from the pose node to the next pose node is the moving edge, which can be obtained from the location part of SLAM algorithm.

After pose estimation, the point cloud data is optimized through non-uniform quadtree to reduce redundant points and improve the resolution and accuracy of the map. Divide the map into multiple quadtree nodes, with each node representing an area. Dynamically adjust the size of nodes based on point cloud density and terrain complexity. $s = s_{\max}/2^n$, where $s$ is the edge length of the node, $s_{\max}$ is the maximum edge length of the node, and $n$ is the number of layers the node is divided into. After each LiDAR scan, insert the newly acquired point cloud data into the quadtree map. Dynamically adjust the resolution of quadtree nodes based on point cloud distribution and terrain features.

$$n = \arg\min_n \left( \frac{N(n)}{s(n)^2} > \gamma \right)$$

(16)

Among them, $N(n)$ is the number of points within the $n$ th layer node, and $\gamma$ is the point density threshold. Based on the semantic information of the obstacles mentioned above, associate the semantic labels of the target obstacles obtained in Neural network-based semantic information extraction for obstacles with the quadtree nodes.

$$L(n) = majority\_vote \left( l_k \,\middle|\, F_k, F_{k+1} \in n \right)$$

(17)

Among them, $L(n)$ is the semantic label of node n, and $l_k$ is the semantic label of point $\{F_k, F_{k+1}\}$. And using the constraint relationship between points and surfaces mentioned above, estimate the pose changes of the tractor.

$$\xi = \arg\min_\xi \left( \sum_{i=1}^{N_e} d_d^2 + \sum_{i=1}^{N_f} d_f^2 \right)$$

(18)

Combine the point cloud data of each frame with the sub map data generated in the previous frame to generate a single sub map. Namely:

$$M_k = M_{k-1} \bigcup P_k$$

(19)

Among them, $M_k$ is the sub map of frame $k$, and $P_k$ is the point cloud data of frame $k$. All sub maps are iteratively updated to generate a global point cloud map, $M_{global} = \bigcup_{k=1}^{N} M_k$. Through the above steps and mathematical expressions, combined with the optimization processing steps of non-uniform quadtree maps, high-precision global point cloud maps can be generated.

**Tractor obstacle avoidance path planning based on A* algorithm and DWA algorithm**

A* algorithm is a widely used path search and graph traversal algorithm, especially suitable for finding the shortest path between two points [18]. In the autonomous obstacle avoidance of mountain tractors, according to the obstacle information in the semantic map, we can identify the obstacles that may affect the tractor's driving, and evaluate their danger level, that is, the cost. A* algorithm can help the tractor find the optimal obstacle avoidance path from the current position to the target position. Set up $v_i$ denotes the coordinate position node in the mountain environment where the tractor is currently located, $u_i$ is the target point. According to the idea of map search, the mountain map is divided into squares with identical specifications. The detailed search steps of the A* algorithm are as follows:

(1) Cost function definition. The A* algorithm evaluates the cost from the tractor start to an arbitrary node $n$ in the mountain map using the cost function $s(n)$. It usually consists of two parts, the actual moving cost $\rho(n)$ of the tractor from the starting point to the node $n$, and the estimated cost $a(n)$ from the node $n$ to the target point $u_i$. Thus, the complete cost function can be expressed as follows:

$$s(n) = \rho(n) + a(n)O$$

[20]

(2) Initialization and node expansion. Put the starting point $v_i$ into the open list (list of nodes to be searched) and the cost of the nodes around its tractor is calculated. These surrounding nodes are added to the open list and sorted according to the cost function.

(3) Select the next node. Select the node with the lowest cost from the open list as the current node and move it to the closed list.

(4) Continue searching. Expand all the neighboring nodes of the current node which are not added to the closed list, calculate their cost and update the open list.

(5) Termination conditions. If the target point $u_i$ is added to the open list, the shortest path from the starting point to the target point is found by backtracking. If the open list is empty but the target point is still not in it, the tractor has no passable obstacle avoidance path.

After the tractor gets the target point, the accessible obstacle avoidance path of the tractor in the mountainous environment is obtained through A* algorithm, and then the track calculation is carried out using DWA algorithm to obtain the corresponding tractor movement strategy. DWA algorithm is a real-time trajectory planning algorithm for mobile tractors in a dynamic environment, which can consider the dynamic constraints and obstacle avoidance requirements of tractors [19]. The detailed operation steps of DWA algorithm are as follows:

(1) Velocity space sampling. Samples within the speed space $(u, v)$ of the tractor to generate a series of possible control commands.

(2) Trajectory simulation. For each sampling point, the movement trajectory of the tractor within a given time interval $\Delta t$ was simulated. Use formula [1] to calculate the new position $(x', y')$ and the new heading angle $\theta'$, the formula is expressed as:

$$\begin{cases} x' = x + v\Delta t \cos(\theta') \\ y' = y + v\Delta t \cos(\theta') \\ \theta' = s(n)\theta + v'\Delta t \end{cases}$$

[21]

Among them, $v$ denotes the linear speed of the tractor. $v'$ denotes the angular velocity of the tractor; $\theta$ indicates the initial heading angle.

(3) Space time constraint setting

In the cost function of the A $*$ algorithm, add the terrain complexity factor $C(n)$:

$$C(n) = \alpha \cdot slope(n) + \beta roughness(n)$$

(22)

Among them, $slope(n)$ is the slope at node $n$, $roughness(n)$ is the roughness at node $n$, and $\alpha$ and $\beta$ are weight coefficients.

Add safety distance constraints to obstacles in the trajectory scoring of DWA algorithm:

$$d_{safe} = \min\left(\sqrt{(x_t - x_o)^2 + (y_t - y_o)^2}\right)$$

(23)

Among them, (xt, yt) are points on the trajectory, and (xo, yo) are the positions of obstacles.

(4) Trajectory rating

Based on the above constraint settings,trajectory scoring. The trajectories of each simulated tractor were scored to evaluate the merits. The scoring function $H(u, v)$ considering deviations from the target heading $e_h$, deviation from the target position $e_d$ and velocity deviations $'e_v$, the formula is expressed as follows:

$$H(u, v) = x'e_h(u, v) + y'e_d(u, v) + \theta'e_v(u, v)$$

[24]

(5) Selection of optimal trajectories. The tractor trajectory with the highest score is selected as the optimal trajectory, and the tractor movement is controlled according to the first control command of this trajectory.

(6) Iteration. In order to ensure that the tractor can reach its destination safely, efficiently and without any obstacles [20], the above process is repeated continuously in order to continuously adjust the robot's trajectory.

The combination of A* algorithm and DWA algorithm can realize autonomous obstacle avoidance path planning of mobile tractor in mountainous environment. Algorithm A is responsible for global path planning and finding the approximate path from the starting point to the target point. DWA algorithm is responsible for local trajectory planning and real-time obstacle avoidance to ensure that the tractor can safely and accurately travel along the global path planned by A* algorithm in complex mountain environment. Through the cooperation of these two algorithms, the autonomous obstacle avoidance ability of mobile tractors in complex mountain environment can be significantly improved.

## Experimental analysis

### Experimental environment

In order to verify the effectiveness of the autonomous obstacle avoidance method proposed in this article for mountain tractors, a representative test area was selected in hilly and mountainous areas, which includes complex terrain such as

slopes of different slopes, ravines of varying depths, and dense tree clusters. The slope changes frequently in hilly and mountainous areas, which poses higher requirements for the tractor's power system and suspension system. Compared with the flat terrain of traditional farmland, mountain tractors need to maintain stability and maneuverability on slopes of different slopes, and the autonomous obstacle avoidance system needs to accurately determine the impact of slope on the driving path. Gullies of varying depths and scattered stones and rocks are common obstacles in mountainous environments. These obstacles not only have irregular shapes, but their positions are also difficult to predict, increasing the complexity and difficulty of obstacle avoidance systems. In contrast, obstacles in traditional farmland are usually more regular and easy to identify. Comprehensively test the tractor's autonomous obstacle avoidance ability in different environments in this complex environment. In the testing area, simulate obstacles such as stones, wooden stakes, and simulated trees according to the experimental design. Selecting the Dongfanghong hilly mountain tractor as the experimental object.

As shown in Fig 4. Ensure that the Dongfanghong tractor is in good working condition, with all mechanical components and electrical systems functioning properly. Install laser radar, GPS receiver, IMU, and other sensors on the Dongfanghong tractor according to the predetermined plan. Lidar should be installed at the front of the tractor to obtain detailed data on the environment ahead; The GPS receiver and IMU should be installed on the top of the tractor or in an appropriate position to reduce signal interference and ensure data accuracy. During installation, use specialized tools and calibration equipment to ensure that the sensor's position and angle are correct. The parameters of the experimental equipment used in this paper are shown in Table 1.

Synchronize the data acquisition time of laser radar, GPS receiver, IMU and other sensors, and calibrate each sensor separately to ensure that the data they acquire can accurately reflect the relative relationship between the tractor and the surrounding environment. Start the laser radar and other sensors, collect and process the mountain environment data, set the starting point and target point of the tractor in the mountain environment, and obtain the obstacle detection results of the tractor when the tractor is running, as shown in Fig 5.

It can be clearly seen from Fig 5 that the smooth surface and irregular shape of rocks in mountain environment are successfully captured by the method in this paper, which effectively distinguishes the difference between rocks and

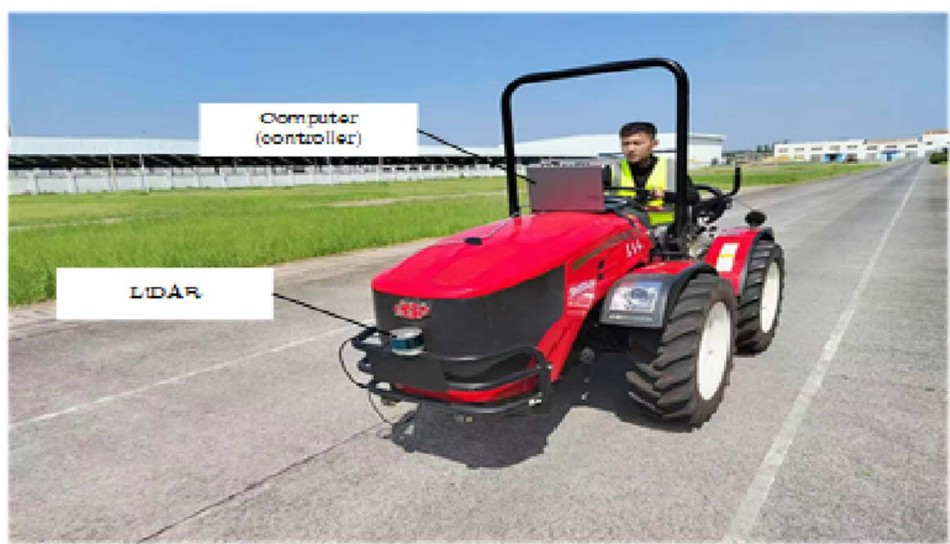

**Fig 4. Dongfanghong tractor.**

**Table 1. Experimental equipment parameters.**

| Equipment name | Model/Specification |
|---|---|
| Laser Radar | RPLIDAR-A1 |
| Tractor platform | Dongfanghong tractor |
| Edge computing Device | NVIDIA Jetson AGX Xavier |
| GPS receiver | Ublox NEO-M8N |
| IMU (Inertial Measurement Unit) | Xsens MTI-30 |
| Sensor bracket and installation accessories | Customized |
| Wireless communication module | Wi-Fi |
| Data storage device | SSD |
| Monitor | Portable monitor |
| RVIZ (Visualization Tool) | Shanhai Whale |

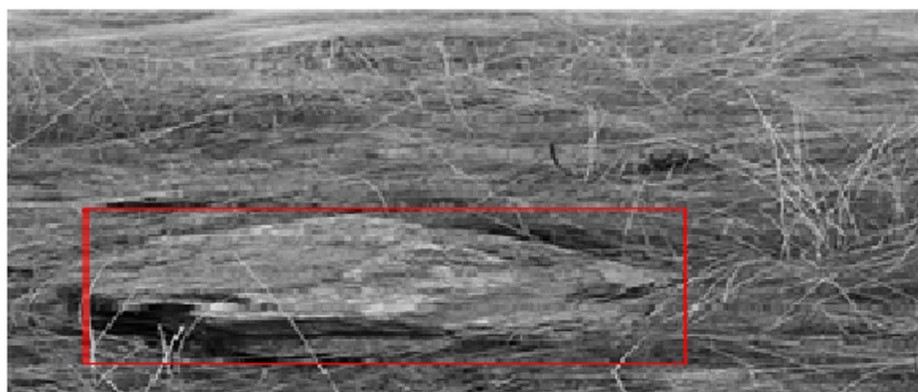

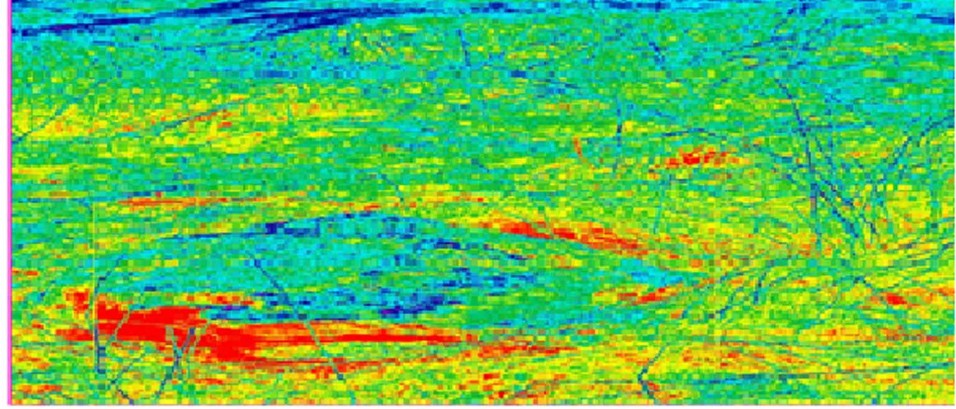

**Fig 5. Obstacle detection results in mountainous environment.**

surrounding environment, accurately identifies and marks a large rock at the center of the photo as an obstacle, proving that even under the interference of complex grassland background and disordered branches and grass leaves, It still maintains a high degree of accuracy and robustness, and shows significant effectiveness in the obstacle detection task in mountain environment.

## Analysis of experimental results

In order to verify the validity of the mountain environment map constructed by the method of this paper, the mountain environment is continuously monitored for a long time in order to obtain the radar data of the mountain environment, which is processed to extract the geomorphological features, and the results of the construction of the mountain environment map by the method of this paper are shown in Fig 6.

As can be seen in Fig 6, the mountain environment map constructed in this paper provides real-time and comprehensive environmental information for mobile devices such as mountain tractors by finely depicting the temperature differences and terrain features in the region. During the traveling process, the tractor can use the highlighted areas in the map to identify potential heat sources or terrain obstacles, such as steep slopes, rock piles, or dense vegetation areas, so as to plan an avoidance path in advance and avoid the risk of collision. In addition, the temperature data in the map can also help identify changes in soil moisture and stability that may result from temperature differences, further reducing the occurrence of accidents such as stuck vehicles.

In order to verify the effectiveness of the method in this paper for estimating the position and attitude of mountain tractors, simulate the position and attitude of tractors in five different scenes, use high-precision measuring equipment to

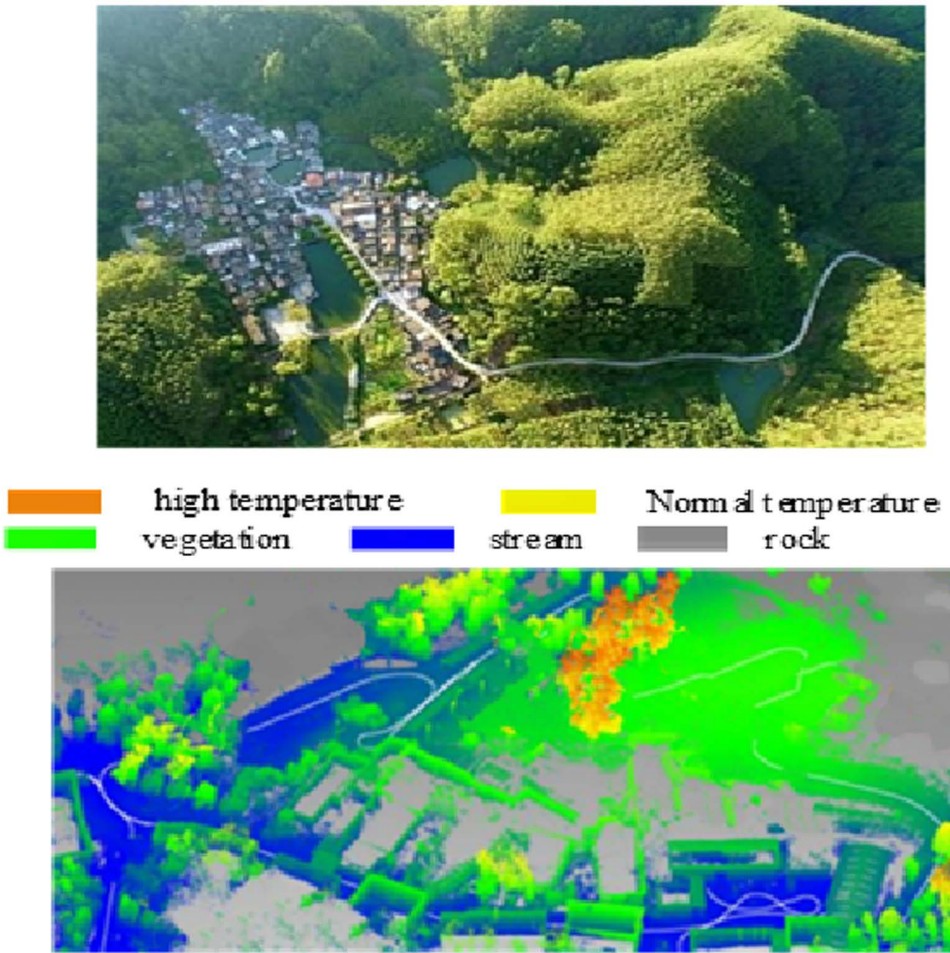

**Fig 6. Construction Results of Mountain Environment Map.**

record the real position and attitude data of tractors, including the angle of the tractor, the distance between the tractor and the x axis, and the distance between the tractor and the y axis. At the same time, apply the GET method FGM method and the method in this paper estimate the position and attitude of the tractor, record the estimated values, and verify the accuracy and reliability of the three methods in the position and attitude estimation of the tractor. The verification results are shown in Table 2.

As can be seen from Table 2, the method in this paper shows high accuracy in angle and distance estimation in all experimental times. Compared with the true value, the estimation error of this method is the smallest, especially the angle estimation is almost close to the true value. In contrast, GET method and FGM method can also give reasonable estimates, but the error is relatively large. At the same time, the estimated value of the method in this paper fluctuates slightly in different experimental times, showing a high stability. It is proved that the profit method in this paper has strong adaptability to different environmental conditions and tractor conditions.

In order to verify the autonomous obstacle avoidance ability of the tractor in the mountain environment, we set the starting point and the end point as the starting and ending positions of the tractor's traveling path, and set a number of irregularly shaped obstacles in the journey, start the tractor, observe and record the tractor's response to the obstacles that may appear along the way from the starting point to the end position, and obtain the tractor's path planning results in Fig 7.

As can be seen in Fig 7, the tractor started from the set starting point and skillfully avoided all the irregularly shaped obstacles along the way, showing high environmental adaptability and intelligent navigation ability. It is especially worth noting that its driving path not only successfully bypasses every obstacle, but also realizes the optimization of the path, ensuring that the distance from the starting point to the end point is the shortest, and there is no unnecessary detouring or turning back in the whole process. It proves the accuracy and efficiency of this method, and also fully proves the reliability and practicality of this method in the complex mountain environment.

In order to further verify the effectiveness of different methods for obstacle avoidance of mountain tractors, a unified starting point and end point are set in the mountain environment, and a number of obstacles with different shapes and positions are arranged between them to simulate the obstacle avoidance challenges in the real work scene. The GET method, FGM method and the method in this paper are applied to the same tractor, and the key performance indicators under each method are recorded, including the driving path length, driving time, number of successful obstacle avoidance,

**Table 2. Tractor pose estimation results using different methods.**

| Number of experiments | Test indicators | True value | GET method | FGM method | Proposed method |
|---|---|---|---|---|---|
| 1 | Angle/° | 123.4 | 122.8 | 122.5 | 123.2 |
|   | X-axis distance/cm | 234.5 | 233.8 | 233.5 | 234 |
|   | Y-axis distance/cm | 345.6 | 345 | 344.8 | 345.4 |
| 2 | Angle/° | 135.7 | 135.2 | 135 | 135.5 |
|   | X-axis distance/cm | 456.7 | 456.1 | 455.9 | 456.3 |
|   | Y-axis distance/cm | 567.8 | 567.3 | 567.1 | 567.6 |
| 3 | Angle/° | 90.1 | 89.8 | 89.6 | 90 |
|   | X-axis distance/cm | 678.9 | 678.2 | 677.9 | 678.5 |
|   | Y-axis distance/cm | 789 | 788.5 | 788.3 | 788.8 |
| 4 | Angle/° | 150.2 | 149.8 | 149.6 | 150 |
|   | X-axis distance/cm | 890.1 | 889.5 | 889.3 | 889.8 |
|   | Y-axis distance/cm | 901.2 | 900.8 | 900.6 | 901 |
| 5 | Angle/° | 67.3 | 67 | 66.8 | 67.2 |
|   | X-axis distance/cm | 101.2 | 100.8 | 100.6 | 101 |
|   | Y-axis distance/cm | 112.3 | 112 | 111.8 | 112.2 |

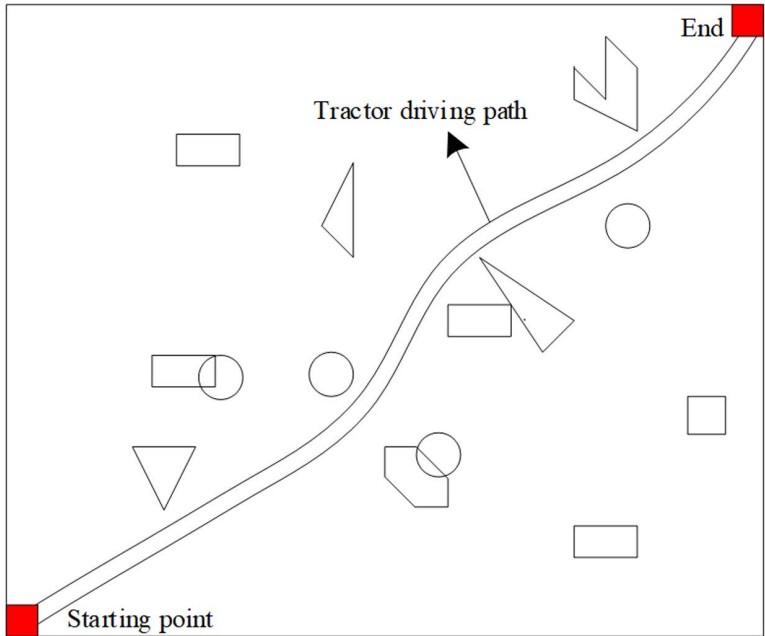

**Fig 7. Obstacle avoidance results of mountain tractors using the method proposed in this article.**

turn back times and driving stability scores. In order to ensure the fairness and repeatability of the experimental results, the same experimental conditions and test standards were used, and repeated tests were conducted for each method for many times to obtain the average data as the final evaluation basis. The comparison results of obstacle avoidance performance of mountain tractors with three methods are shown in Table 3.

It can be seen from Table 3 that the method in this paper performs well in two key indicators: path length and travel time. Compared with GET method and FGM method, the path length is shortened by 2.8% and 5.5% respectively, and the travel time is significantly reduced by 12.5% and 14.8%, which proves that the method in this paper has obvious advantages in path planning and optimization, and can effectively shorten the travel distance and improve operating efficiency. In terms of the number of successful obstacle avoidance, the method in this paper has reached 5, significantly higher than the GET method and the FGM method, which proves that the method in this paper is more flexible and effective in dealing with diverse obstacles in complex mountain environment, and has stronger obstacle avoidance ability. In terms of smoothness score, the method in this paper gets a full score of 10 points, while GET method and FGM method get 7 points and 8 points respectively, which fully proves that the method in this paper has significant effects on improving the driving smoothness of tractors, which helps to reduce mechanical wear, improve operation quality and enhance driver comfort.

**Table 3. Comparison results of obstacle avoidance performance of mountain tractors using different methods.**

| Evaluating indicator | GET method | FGM method | Proposed method |
|---|---|---|---|
| Road length/m | 131.6 | 135.3 | 127.8 |
| Travel time/m | 182.5 | 187.4 | 159.7 |
| Number of successfully avoided obstacles/each | 3 | 3 | 5 |
| Number of turns/times | 2 | 3 | 0 |
| Stability score (over 10 points) | 7 | 8 | 10 |

At the same time, the method in this paper did not turn back during the experiment. In contrast, the GET method and the FGM method turned back twice and three times respectively, indicating that the method in this paper is more accurate and efficient in the path planning and execution process, can effectively avoid unnecessary turns and detours, and further improve the work efficiency.

To further validate the applicability of the design method and test the power consumption and delay tolerance in different distance environments, a comparative test analysis was conducted between the reinforcement learning based vehicle control algorithm in reference 9 and the Bezier curve based vehicle control algorithm in reference 10. The results are shown below.

According to the data in Table 4, the power consumption of the Proposed method is significantly lower than the other two algorithms at a distance of 500 meters. The power consumption of the Proposed method is 0.2 kWh, while the algorithms based on reinforcement learning and Bezier curve are 0.86 kWh and 0.77 kWh, respectively. This indicates that the proposed method performs the best in terms of power consumption and has higher energy efficiency. And the delay tolerance of the Proposed method * * is significantly lower than the other two algorithms. At a distance of 500 meters, the delay tolerance of the Proposed method is 0.12 seconds, while the reinforcement learning based algorithm and the Bezier curve based algorithm are 1.5 seconds and 1.4 seconds, respectively. This indicates that the proposed method performs the best in terms of delay tolerance, with higher real-time performance and response speed.

To further verify the safety performance of the design method, a minimum obstacle avoidance distance test analysis was conducted, and the results are shown in the following figure (Fig 8).

Based on the above results, it can be seen that the minimum obstacle avoidance distance of the algorithm in this paper is always around 8m and relatively stable, while the minimum obstacle avoidance distance of the vehicle control algorithm based on reinforcement learning fluctuates greatly between 1-5m and is unstable; The minimum obstacle avoidance distance of the vehicle control algorithm based on the Beizer curve hovers between 0.6 and 4 meters, which is too small and poses a certain degree of danger, and the fluctuation is greater than that of the vehicle control algorithm based on reinforcement learning. This indicates that the security performance of the algorithm in this article is relatively good.

To verify whether the method proposed in this article is applicable to embedded devices, a memory usage analysis was conducted, and the results are shown in the following Table 5:

The memory usage of the proposed method is significantly lower than the other two algorithms. At 5 days, the proposed method had a memory usage rate of 8%, while the reinforcement learning based algorithm and the Bezier curve based algorithm had memory usage rates of 11% and 10%, respectively. As time goes on, the memory usage of the proposed method grows relatively steadily. This indicates that the proposed method performs the best in terms of memory usage and has higher memory efficiency. Can effectively embed devices.

Table 4. Results of comparing power consumption and minimum obstacle avoidance distance at different distances.

| distance/m | A vehicle control algorithm based on reinforcement learning | | Vehicle control algorithm based on Bezer curve | | Proposed method | |
|---|---|---|---|---|---|---|
| | Power consumption/kWh | Delay tolerance/s | Power consumption/kWh | Delay tolerance/s | Power consumption/kWh | Delay tolerance/s |
| 500 | 0.86 | 1.5 | 0.77 | 1.4 | 0.2 | 0.12 |
| 1000 | 0.99 | 1.9 | 0.86 | 1.9 | 0.39 | 0.24 |
| 1500 | 1.20 | 2.1 | 0.92 | 2.1 | 0.61 | 0.33 |
| 2000 | 1.35 | 2.5 | 1.32 | 2.3 | 0.77 | 0.41 |
| 2500 | 1.41 | 5.9 | 1.50 | 2.5 | 0.95 | 0.52 |

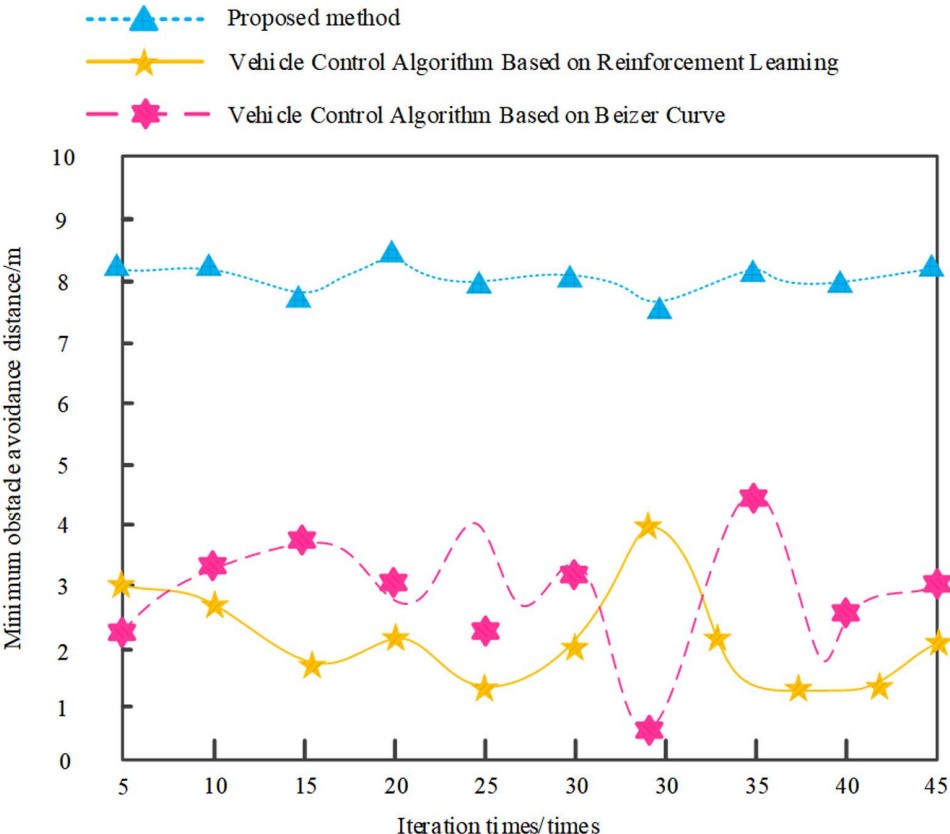

**Fig 8. Minimum obstacle avoidance distance test results.**

**Table 5. Comparison results of memory occupancy at different times.**

| Time/ day | Vehicle control algorithm based on reinforcement learning/% | Vehicle control algorithm based on Bezer curve/% | Proposed method/% |
|---|---|---|---|
| 5 | 11 | 10 | 8 |
| 10 | 15 | 15 | 9 |
| 15 | 18 | 16 | 10 |
| 20 | 19 | 17 | 10 |
| 25 | 21 | 19 | 12 |
| 30 | 22 | 20 | 14 |

## Conclusion

This paper studies the autonomous obstacle avoidance method of mountain tractors based on semantic neural network and laser SLAM technology, and comprehensively verifies the excellent performance of this method through experiments. The experimental conclusions are as follows. This method can sensitively capture and distinguish various obstacles in the complex mountain environment, and also realize the fine construction and real-time update of the environment map. It can intelligently plan a safe and efficient driving path, effectively shorten the driving distance and reduce the time consumption. The driving smoothness of the tractor is significantly enhanced, the mechanical wear is reduced through precise control, the operation quality is improved, and the driver's comfort and satisfaction are greatly improved. In the process of

experiment, the efficient path execution and precise control capabilities demonstrated by this method further confirmed its reliability and practicability in the complex mountain environment.

## Supporting information

**S1 File. Data for parameters.**
(XLSX)

## Acknowledgements

Thank you for the experimental assistance provided by the School of Vehicle and Transportation Engineering, Henan University of Science and Technology.

## Author contributions

**Conceptualization:** Ningjie Chang, Liyou Xu.

**Funding acquisition:** Xianghai Yan.

**Methodology:** Ningjie Chang, Yiwei Wu.

**Software:** Bingxin Chen.

**Supervision:** Liyou Xu.

**Writing – original draft:** Ningjie Chang.

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
