## [Decision Letter · Decision Letter 0]

27 Feb 2025

PONE-D-25-03307Research on Autonomous Obstacle Avoidance of Mountainous Tractors Based on Semantic Neural Network and Laser SLAMPLOS ONE

Dear Dr. Xu,

Thank you for submitting your manuscript to PLOS ONE. After careful consideration, we feel that it has merit but does not fully meet PLOS ONE’s publication criteria as it currently stands. Therefore, we invite you to submit a revised version of the manuscript that addresses the points raised during the review process. This paper proposes a research on autonomous obstacle avoidance of mountainous tractors based on semantic neural network and laser SLAM. But revisions are needed in terms of comparison studies, writing, table and figure formulation and literature review comprehensiveness. 

We look forward to receiving your revised manuscript.

Kind regards,

Lei Zhang, PhD

Academic Editor

PLOS ONE

Journal Requirements:

Key Research and Development Project of Henan Province, 231111112600; Natural Science Foundation of Henan Province, 242300420370; Training Program for Young Backbone Teachers in Undergraduate Universities in Henan Province, 2024GGJS051; Heluo Youth Talent Support Project, 2024 HLTJ03; Henan Province University Science and Technology Innovation Team Support Program Project, 24IRTSTHN029; Key Research Project Plan for Higher Education Institutions in Henan Province, 25B460004

5. We note that Figure 4 includes an image of a [patient / participant / in the study].

Please respond by return e-mail with an amended manuscript. We can upload this to your submission on your behalf.

If you are unable to obtain consent from the subject of the photograph, please either instruct us to remove the figure or supply a replacement figure by return e-mail for which you hold the relevant copyright permissions and subject consents. In some cases, you may need to specify in the text that the image used in the figure is not the original image used in the study, but a similar image used for illustrative purposes only. We can make any changes on your behalf.

Additional Editor Comments :

This paper proposes a research on autonomous obstacle avoidance of mountainous tractors based on semantic neural network and laser SLAM. But revisions are needed in terms of comparison studies, writing, table and figure formulation and literature review comprehensiveness. Pay attention to that Reviewer 1's comments are presented in the attached file.

Reviewers' comments:

Reviewer's Responses to Questions

**Comments to the Author**

1. Is the manuscript technically sound, and do the data support the conclusions?

Reviewer #1: Yes

Reviewer #2: Yes

2. Has the statistical analysis been performed appropriately and rigorously? 

Reviewer #1: No

Reviewer #2: Yes

3. Have the authors made all data underlying the findings in their manuscript fully available?

Reviewer #1: No

Reviewer #2: No

4. Is the manuscript presented in an intelligible fashion and written in standard English?

Reviewer #1: Yes

Reviewer #2: Yes

5. Review Comments to the Author

Reviewer #1: Restructure the methodology section to clarify technical details; Expand the experimental section (dynamic scenarios, multi-sensor comparisons ); Add feasibility analysis for practical deployment.

Reviewer #2: 1. The experimental environment was mentioned in the paper, but the details were insufficient. Suggest providing a detailed description of the terrain features of the testing area, such as slope, soil type, vegetation density, etc., as well as the types and arrangement of obstacles. This will help other researchers replicate the experiment and validate the results.

2. Please carefully check and confirm whether the figures, tables, etc. in the article conform to the format

3. In addition to comparing with GET and FGM methods, it is recommended to add comparisons with other advanced methods (such as deep learning based obstacle avoidance methods) to more comprehensively evaluate the performance of the proposed method.

4. Some variables that appear for the first time in the formula have not been explained in the main text, such as formula 5. It is recommended to add explanations.

5. Increase the review of existing research, especially the latest research progress related to semantic neural networks and laser SLAM technology. This will help readers better understand the innovation and background of the proposed method. Recommended quotes include "Enhancing Vehicle Re identification by Pair flexible Pose Guided Vehicle Image Synthesis" and "Deep Transfer Learning for Intelligent Vehicle Perception: A Survey"

6. Please ensure that the charts in the paper are clear and easy to read, and that the captions are detailed and accurate. For complex charts, it is recommended to add necessary explanations and clarifications

7. It is recommended to add a comparative analysis with the proposed method when describing existing research methods, clearly pointing out the shortcomings of the existing methods and the areas for improvement of the proposed method. This will help highlight the research contributions and innovative points of the paper.

6. PLOS authors have the option to publish the peer review history of their article (what does this mean? ). If published, this will include your full peer review and any attached files.

**Do you want your identity to be public for this peer review?** For information about this choice, including consent withdrawal, please see our Privacy Policy .

Reviewer #1: No

Reviewer #2: No

---

## [Author Response · Author response to Decision Letter 1]

7 Apr 2025

1. The introduction does not clearly articulate the unique challenges of autonomous obstacle avoidance for mountain tractors. A detailed scenario analysis is needed to highlight the differences from traditional farmland environments.

Response: Provide detailed scenarios following your recommendations to clarify the differences from traditional farmland environments, such as:

In order to verify the effectiveness of the autonomous obstacle avoidance method proposed in this article for mountain tractors, a representative test area was selected in hilly and mountainous areas, which includes complex terrain such as slopes of different slopes, ravines of varying depths, and dense tree clusters. The slope changes frequently in hilly and mountainous areas, which poses higher requirements for the tractor's power system and suspension system. Compared with the flat terrain of traditional farmland, mountain tractors need to maintain stability and maneuverability on slopes of different slopes, and the autonomous obstacle avoidance system needs to accurately determine the impact of slope on the driving path. Gullies of varying depths and scattered stones and rocks are common obstacles in mountainous environments. These obstacles not only have irregular shapes, but their positions are also difficult to predict, increasing the complexity and difficulty of obstacle avoidance systems. In contrast, obstacles in traditional farmland are usually more regular and easy to identify. Comprehensively test the tractor's autonomous obstacle avoidance ability in different environments in this complex environment. In the testing area, simulate obstacles such as stones, wooden stakes, and simulated trees according to the experimental design. Selecting the John Deere 6125R hilly mountain tractor as the experimental object,

2. The mathematical derivation of plane feature matching in Equations (8)-(10) is overly simplified, with no explanation of how noise in point cloud registration is addressed. It is recommended to refer to the GEITS paper "A non-uniform quadtree map building method including dead-end semantics extraction" (DOI:10.1016/j.geits.2023.100071) for map construction methods.

Response: Following your suggestion, we have added a feature matching process for the non-uniform quadtree map construction method, such as:

Firstly, remove outliers in the point cloud that are too far from the mean. , Among them, is the -th point in the point cloud, and is the value of the point cloud. If (threshold), it is removed and changed to, and the data is denoised according to the above point cloud filtering.

After pose estimation, the point cloud data is optimized through non-uniform quadtree to reduce redundant points and improve the resolution and accuracy of the map. Divide the map into multiple quadtree nodes, with each node representing an area. Dynamically adjust the size of nodes based on point cloud density and terrain complexity. , where is the edge length of the node, is the maximum edge length of the node, and is the number of layers the node is divided into. After each LiDAR scan, insert the newly acquired point cloud data into the quadtree map. Dynamically adjust the resolution of quadtree nodes based on point cloud distribution and terrain features.

16

Among them, is the number of points within the th layer node, and is the point density threshold. Based on the semantic information of the obstacles mentioned above, associate the semantic labels of the target obstacles obtained in section 2.2 with the quadtree nodes.

17

Among them, is the semantic label of node n, and is the semantic label of point . And using the constraint relationship between points and surfaces mentioned above, estimate the pose changes of the tractor.

18

Combine the point cloud data of each frame with the sub map data generated in the previous frame to generate a single sub map. Namely:

19

Among them, is the sub map of frame , and is the point cloud data of frame . All sub maps are iteratively updated to generate a global point cloud map, . Through the above steps and mathematical expressions, combined with the optimization processing steps of non-uniform quadtree maps, high-precision global point cloud maps can be generated.

3. Only GET and FGM algorithms are compared, without including mainstream methods like RRT* or deep reinforcement learning. It is recommended to reference the baseline methods in the GEITS paper "A review on reinforcement learning-based highway autonomous vehicle control" (DOI:10.1016/j.geits.2024.100156).

Response: We have supplemented the above references as per your suggestion and conducted comparative analysis with them, such as:

Irshayyid A et al. first reviewed the different traffic scenarios discussed in the literature [9], and then conducted a comprehensive review of DRL technology, such as the state representation method for capturing the interactive dynamics required for safe and efficient merging, and the reward formula for managing key indicators such as safety, efficiency, comfort, and adaptability. The insights from this review can guide future research towards realizing the potential of DRL in complex traffic automation under uncertainty. A dynamic adaptive trajectory planning method based on Bezier curve is proposed.

To further validate the applicability of the design method and test the power consumption and delay tolerance in different distance environments, a comparative test analysis was conducted between the reinforcement learning based vehicle control algorithm in reference [9] and the Bezier curve based vehicle control algorithm in reference [10]. The results are shown below.

Table 4 Results of comparing power consumption and minimum obstacle avoidance distance at different distances

distance /m A vehicle control algorithm based on reinforcement learning Vehicle control algorithm based on Bezer curve Proposed method

Power consumption/kWh Delay tolerance/s Power consumption/kWh Delay tolerance/s Power consumption/kWh Delay tolerance/s

500 0.86 1.5 0.77 1.4 0.2 0.12

1000 0.99 1.9 0.86 1.9 0.39 0.24

1500 1.20 2.1 0.92 2.1 0.61 0.33

2000 1.35 2.5 1.32 2.3 0.77 0.41

2500 1.41 5.9 1.50 2.5 0.95 0.52

According to the data in Table 4, the power consumption of the Proposed method is significantly lower than the other two algorithms at a distance of 500 meters. The power consumption of the Proposed method is 0.2 kWh, while the algorithms based on reinforcement learning and Bezier curve are 0.86 kWh and 0.77 kWh, respectively. This indicates that the proposed method performs the best in terms of power consumption and has higher energy efficiency. And the delay tolerance of the Proposed method * * is significantly lower than the other two algorithms. At a distance of 500 meters, the delay tolerance of the Proposed method is 0.12 seconds, while the reinforcement learning based algorithm and the Bezier curve based algorithm are 1.5 seconds and 1.4 seconds, respectively. This indicates that the proposed method performs the best in terms of delay tolerance, with higher real-time performance and response speed.

4. The coordination mechanism between A* and DWA is vaguely described, with no explanation of how global paths are dynamically updated to adapt to environmental changes. It is recommended to refer to the spatiotemporal constraint optimization methods in the GEITS paper "Spatiotemporal-restricted A∗ algorithm as a support for lane-free traffic" (DOI:10.1016/j.geits.2024.100159).

Response: We have referred to relevant papers as per your suggestion, supplemented with spatiotemporal constraints and other conditions to adapt to environmental changes, such as:

(3) Space time constraint setting

In the cost function of the A * algorithm, add the terrain complexity factor :

22

Among them, is the slope at node , is the roughness at node , and and are weight coefficients.

Add safety distance constraints to obstacles in the trajectory scoring of DWA algorithm:

23

Among them, (xt, yt) are points on the trajectory, and (xo, yo) are the positions of obstacles.

(4) Trajectory rating:

Based on the above constraint settings,trajectory scoring. The trajectories of each simulated tractor were scored to evaluate the merits. The scoring function considering deviations from the target heading , deviation from the target position and velocity deviations , the formula is expressed as follows:

(24)

5. Key metrics such as inference speed (FPS) and memory usage of the lightweight network are missing, making it impossible to prove its suitability for embedded devices.

Response: Complthe comparative analysis following your recommendations, such as:

To verify whether the method proposed in this article is applicable to embedded devices, a memory usage analysis was conducted, and the results are shown in the following table:

Table 4 Comparison results of memory occupancy at different times

Time / day Vehicle control algorithm based on reinforcement learning /% Vehicle control algorithm based on Bezer curve/% Proposed method/%

5 11 10 8

10 15 15 9

15 18 16 10

20 19 17 10

25 21 19 12

30 22 20 14

The memory usage of the proposed method is significantly lower than the other two algorithms. At 5 days, the proposed method had a memory usage rate of 8%, while the reinforcement learning based algorithm and the Bezier curve based algorithm had memory usage rates of 11% and 10%, respectively. As time goes on, the memory usage of the proposed method grows relatively steadily. This indicates that the proposed method performs the best in terms of memory usage and has higher memory efficiency. Can effectively embed devices.

6. The data availability statement claims "data is within the manuscript," but no point cloud dataset or code links are provided, violating reproducibility principles. Data should be uploaded to a public platform.

Response: Regarding the issue you mentioned that the point cloud dataset and code links were not provided, we would like to clarify that some of the data involved in this study do contain proprietary information, which cannot be fully uploaded to public platforms. The publicly available data has been provided with relevant information in the article to ensure the transparency and verifiability of the research.

7. The system's power consumption and its impact on tractor endurance, especially in mountainous areas without charging facilities, are not discussed. This issue is critical and should be addressed.

Response: The power consumption test analysis of this system has been supplemented by following your recommendations, such as:

To further validate the applicability of the design method and test the power consumption and delay tolerance in different distance environments, a comparative test analysis was conducted between the reinforcement learning based vehicle control algorithm in reference [9] and the Bezier curve based vehicle control algorithm in reference [10]. The results are shown below.

Table 4 Results of comparing power consumption and minimum obstacle avoidance distance at different distances

distance /m A vehicle control algorithm based on reinforcement learning Vehicle control algorithm based on Bezer curve Proposed method

Power consumption/kWh Delay tolerance/s Power consumption/kWh Delay tolerance/s Power consumption/kWh Delay tolerance/s

500 0.86 1.5 0.77 1.4 0.2 0.12

1000 0.99 1.9 0.86 1.9 0.39 0.24

1500 1.20 2.1 0.92 2.1 0.61 0.33

2000 1.35 2.5 1.32 2.3 0.77 0.41

2500 1.41 5.9 1.50 2.5 0.95 0.52

According to the data in Table 4, the power consumption of the Proposed method is significantly lower than the other two algorithms at a distance of 500 meters. The power consumption of the Proposed method is 0.2 kWh, while the algorithms based on reinforcement learning and Bezier curve are 0.86 kWh and 0.77 kWh, respectively. This indicates that the proposed method performs the best in terms of power consumption and has higher energy efficiency. And the delay tolerance of the Proposed method * * is significantly lower than the other two algorithms. At a distance of 500 meters, the delay tolerance of the Proposed method is 0.12 seconds, while the reinforcement learning based algorithm and the Bezier curve based algorithm are 1.5 seconds and 1.4 seconds, respectively. This indicates that the proposed method performs the best in terms of delay tolerance, with higher real-time performance and response speed.

8. Safety is only measured by "comfort score," without introducing quantitative metrics from agricultural machinery safety standards (e.g., ISO 25119), such as minimum obstacle avoidance distance or emergency braking response time.

Response: The minimum obstacle avoidance distance test analysis has been supplemented by following your recommendations, such as:

To further verify the safety performance of the design method, a minimum obstacle avoidance distance test analysis was conducted, and the results are shown in the following figure.

Figure 8 Minimum obstacle avoidance distance test results

Based on the above results, it can be seen that the minimum obstacle avoidance distance of the algorithm in this paper is always around 8m and relatively stable, while the minimum obstacle avoidance distance of the vehicle control algorithm based on reinforcement learning fluctuates greatly between 1-5m and is unstable; The minimum obstacle avoidance distance of the vehicle control algorithm based on the Beizer curve hovers between 0.6 and 4 meters, which is too small and poses a certain degree of danger, and the fluctuation is greater than that of the vehicle control algorithm based on reinforcement learning. This indicates that the security performance of the algorithm in this article is relatively good.

9. In distributed control scenarios, communication delays between LiDAR and computing units may affect real-time performance. Delay tolerance testing should be added.

Response: The delay tolerance test analysis has been supplemented by following your recommendations, such as:

To further validate the applicability of the design method and test the power consumption and delay tolerance in different distance environments, a comparative test analysis was conducted between the reinforcement learning based vehicle control algorithm in reference [9] and the Bezier curve based vehicle control algorithm in reference [10]. The results are shown below.

Table 4 Results of comparing power consumption and minimum obstacle avoidance distance at different distances

distance /m A vehicle control algorithm based on reinforcement learning Vehicle control algorithm based on Bezer curve Proposed method

Power consumption/kWh Delay tolerance/s Power consumption/kWh Delay tolerance/s Power consumption/kWh Delay tolerance/s

500 0.86 1.5 0.77 1.4 0.2 0.12

1000 0.99 1.9 0.86 1.9 0.39 0.24

1500 1.20 2.1 0.92 2.1 0.61 0.33

2000 1.35 2.5 1.32 2.3 0.77 0.41

2500 1.41 5.9 1.50 2.5 0.95 0.52

According to the data in Table 4, the power consumption of the Proposed method is significantly lower than the other two algorithms at a distance of 500 meters. The power consumption of the Proposed method is 0.2 kWh, while the algorithms based on reinforcement learning and Bezier curve are 0.86 kWh and 0.77 kWh, respectively. This indicates that the proposed method performs the best in terms of power consumption and has higher energy efficiency. And the delay tolerance of the Proposed method * * is significantly lower than the other two algorithms. At a distance of 500 meters, the delay tolerance of the Proposed method is 0.12 seconds, while the reinforcement learning based algorithm and the Bezier curve based algorithm are 1.5 seconds and 1.4 seconds, respectively. This indicates that the proposed method performs the best in terms of delay tolerance, with higher real-time performance and response speed.

10. Some references (e.g., [9] on laser SLAM) are from before 2023 and you can cite the latest GEITS paper "Deep transfer learning for intelligent vehicle perception: A survey" (DOI:

---

## [Decision Letter · Decision Letter 1]

11 Apr 2025

Research on Autonomous Obstacle Avoidance of Mountainous Tractors Based on Semantic Neural Network and Laser SLAM

PONE-D-25-03307R1

Dear Dr. Xu,

We’re pleased to inform you that your manuscript has been judged scientifically suitable for publication and will be formally accepted for publication once it meets all outstanding technical requirements.

Kind regards,

Lei Zhang, PhD

Academic Editor

PLOS ONE

Additional Editor Comments (optional):

The revised version is publishable.

Reviewers' comments:

Reviewer's Responses to Questions

**Comments to the Author**

1. If the authors have adequately addressed your comments raised in a previous round of review and you feel that this manuscript is now acceptable for publication, you may indicate that here to bypass the “Comments to the Author” section, enter your conflict of interest statement in the “Confidential to Editor” section, and submit your "Accept" recommendation.

Reviewer #1: All comments have been addressed

Reviewer #2: (No Response)

2. Is the manuscript technically sound, and do the data support the conclusions?

Reviewer #1: Yes

Reviewer #2: (No Response)

3. Has the statistical analysis been performed appropriately and rigorously? 

Reviewer #1: Yes

Reviewer #2: (No Response)

4. Have the authors made all data underlying the findings in their manuscript fully available?

Reviewer #1: Yes

Reviewer #2: (No Response)

5. Is the manuscript presented in an intelligible fashion and written in standard English?

Reviewer #1: Yes

Reviewer #2: (No Response)

6. Review Comments to the Author

Reviewer #1: This paper presents an autonomous obstacle avoidance method for mountainous tractors using semantic neural networks and laser SLAM. It integrates LiDAR-based environment scanning, a pruned YOLOv3 model for obstacle detection, and A* with DWA algorithms for path planning. Experimental results demonstrate improved accuracy, efficiency, and safety over traditional methods, with lower power consumption and memory usage. However, the study lacks comparisons with advanced algorithms like RRT* and detailed validation under dynamic conditions, necessitating further refinement for real-world deployment.

Reviewer #2: (No Response)

7. PLOS authors have the option to publish the peer review history of their article (what does this mean? ). If published, this will include your full peer review and any attached files.

**Do you want your identity to be public for this peer review?** For information about this choice, including consent withdrawal, please see our Privacy Policy .

Reviewer #1: No

Reviewer #2: No

---

## [Editor Report · Acceptance letter]

PONE-D-25-03307R1

PLOS ONE

Dear Dr. Xu,

I'm pleased to inform you that your manuscript has been deemed suitable for publication in PLOS ONE. Congratulations! Your manuscript is now being handed over to our production team.

Kind regards,

on behalf of

Dr. Lei Zhang

Academic Editor

PLOS ONE